# Lower Functional and Proportional Characteristics of Cord Blood Treg of Male Newborns Compared with Female Newborns

**DOI:** 10.3390/biomedicines9020170

**Published:** 2021-02-09

**Authors:** Viktor Černý, Olga Novotná, Petra Petrásková, Kateřina Hudcová, Kristýna Boráková, Ludmila Prokešová, Libuše Kolářová, Jiří Hrdý

**Affiliations:** 1Institute of Immunology and Microbiology, First Faculty of Medicine, Charles University, 121 08 Prague, Czech Republic; viktor.cerny@lf1.cuni.cz (V.Č.); olga.novotna@lf1.cuni.cz (O.N.); petra.petraskova@lf1.cuni.cz (P.P.); macholdk@gmail.com (K.H.); lprok@lf1.cuni.cz (L.P.); libuse.kolarova@lf1.cuni.cz (L.K.); 2Institute for the Care of Mother and Child, 147 00 Prague, Czech Republic; borakova@upmd.cz

**Keywords:** regulatory T cells, allergy, cord blood, induced Treg, natural Treg, flow cytometry, Helios, epigenetics, TSDR

## Abstract

Understanding the early events involved in the induction of immune tolerance to harmless environmental antigens and microbiota compounds could reveal potential targets for allergic disease therapy or prevention. Regulatory T cells (Treg), particularly induced Treg (iTreg), are crucial for the induction and maintenance of tolerance against environmental antigens including allergens. A decrease in the number and/or function of Treg or iTreg could represent an early predictor of allergy development. We analyzed proportional and functional properties of Treg in the cord blood of children of allergic mothers (neonates at high risk of allergy development) and healthy mothers (neonates with relatively low risk of allergy development). We observed a higher number of induced Treg in the cord blood of females compared to males, suggesting an impaired capacity of male immunity to set up tolerance to allergens, which could contribute to the higher incidence of allergy observed in male infants. The decreased proportion of iTreg in cord blood compared with maternal peripheral blood documents the general immaturity of the neonatal immune system. We observed a positive correlation in the demethylation of the Treg-specific demethylated region (TSDR) and the proportion of Treg in cord blood. Our data suggest that immaturity of the neonatal immune system is more severe in males, predisposing them to increased risk of allergy development.

## 1. Introduction

Allergic diseases are a highly diverse, multifactorial group of immunological disorders, characterized by unwarranted immune reactivity to and/or failure of tolerance to relatively innocuous exogenous antigens, i.e., allergens. The underlying cause of allergy is a dysregulation of the fine balance among various branches of the immune system which are orchestrated by specialized subpopulations of CD4^+^ T cells (e.g., Th1, Th2, Th9, Treg, Th17, Th22). The roots of this dysregulation can be tracked to the prenatal period, when developing fetal immunity is maintained under Th2 bias in order to prevent harmful reactivity against antigenically foreign maternal determinants [1]. This Th2 bias needs to be postnatally retracted and a new balance among immune response branches must be established in order to avoid Th2- and immunoglobulin E (IgE)-mediated sensitization and hyper-responsiveness toward innocuous environmental antigens, which constitute the most common form of allergy.

Early postnatal rebalancing of immune system branches is regulated by many environmental and intrinsic factors, including exposure to microbiota [2]. Central among the intrinsic factors constituting and maintaining the proper balance among branches of the immune response are regulatory T cells (Treg), a key immunomodulatory population also responsible for induction and maintenance of tolerance towards autoantigens and harmless environmental antigens, including allergens [3].

In humans, Treg are characterized as CD4^+^CD25^+/high^CD127^low^ cells which express FOXP3 as a lineage-specific transcription factor. The crucial importance of FOXP3^+^ Treg can be demonstrated by severe syndromes caused by FOXP3 deficiency, which include multiple symptoms of autoimmunity and allergy, both in humans [4,5] and in mice [6]. MFI of FOXP3 has, in fact, been described to correlate with the Treg suppressive function [7]. Other Treg characteristics include high dependence on exogenously produced IL-2, production of typical suppressive cytokines IL-10, TGF-β and IL-35 [8] and expression of various other cell surface and intracellular markers, associated with Treg function and/or their lineage stability, many of which are useful for indirectly assessing the Treg functional capacity. Commonly measured Treg cell surface markers include CTLA-4, PD-1 and LAG-3, utilized in contact-dependent inhibition of effector immune responses [8].

Lately, epigenetic regulation of Treg has been proposed as a key mechanism which plays a role both complementary to and independent from that of FOXP3 in inducing and maintaining Treg function and lineage stability [9]. A characteristic demethylation pattern of specific conserved noncoding sequences (CNS) located within the FOXP3 gene has been identified as the crucial epigenetic determinant of the Treg phenotype, stability and function [10]. The most important of these hypomethylated areas is the widely evolutionarily conserved CNS2, also called the Treg-specific demethylated region (TSDR) [11]. The biological importance of the TSDR is illustrated by pronounced autoimmune and allergic symptoms accompanying aberrant epigenetic control of an otherwise intact FOXP3 locus [12]; indeed, it has been demonstrated that single-nucleotide polymorphisms associated with various autoimmune diseases are significantly enriched in noncoding areas critically involved in epigenetic control of Treg, including the TSDR [13]. Additionally, TSDR demethylation has been described as a marker of effective antigen-specific immunotherapy of allergy [14,15] and induction of oral tolerance [16].

It has been postulated that Treg can be broadly divided into two major subpopulations—natural Treg (nTreg), which are generated during thymic negative selection and which predominantly express T cell receptors (TCR) specific toward autoantigens and play a major role in preventing autoimmune disorders [17]; and induced Treg (iTreg), which arise in peripheral tissues under tolerogenic conditions (especially in mucosal environments rich in TGF-β) and which possess TCR specificity mainly toward innocuous exoantigens and play a key role in controlling tolerance towards harmless external antigens including allergens [18,19]. The two broad populations have yet to be conclusively distinguished based on the expression of easily measurable markers, although the expression of Ikaros family transcription factor Helios has been proposed to be specific for nTreg but not iTreg by some studies [20]. Peripherally induced iTreg, and particularly *in vitro* generated iTreg, have been described by various studies as less stable than nTreg, especially under inflammatory conditions [21,22]. This corresponds to lower levels of demethylation of the TSDR in iTreg [23,24,25] and can be improved by implementation of specific protocols that promote DNA demethylation, such as addition of vitamin C [26] or retinoic acid [27] or withdrawal of CD28 signaling during Treg induction [28].

Due to the globally high medical, economic and social burden allergic diseases continue to pose, many studies have been and are being conducted to assess the suitability of various immunological parameters as factors useful for prediction of individual risk of allergy development. Identification of such factors could enable clinical testing and early introduction of appropriate preventive measures or at least more timely intervention. Cord blood is a particularly promising biological material for similar studies as it is non-invasively and readily available at the moment of birth. Nevertheless, despite the progress in our understanding of the early etiopathology of allergy development, few reliable predictors of individual risk have been conclusively identified so far. Various factors have been studied without much success, including cord blood IgE [29,30], levels of IL-10, TGF-β and/or Th1- and Th2-related cytokines in cord blood [31,32] and neonatal peripheral blood plasma [33], as well as the reactivity of cord blood-derived cells to diverse stimuli under various conditions [34,35,36,37,38]. So far, parental allergy status, especially maternal or biparental allergy, seems to be the most reliable predictor of risk for allergy development at an individual level [38,39]. Further research in this area is therefore warranted.

In our study, we aimed to compare proportional and selected functional characteristics of total Treg, nTreg and iTreg as well as the TSDR demethylation status of Treg in the cord blood of children of allergic mothers (with high risk of allergy development) and newborns of healthy mothers (with lower risk of allergy development).

## 2. Materials and Methods

### 2.1. Subjects and Sample Collection

The study included healthy (n = 118) and allergic (n = 108) mothers with physiological pregnancies who delivered children via cesarean section at full term in the Institute for the Care of Mother and Child in Prague, Czech Republic. The allergy status of the mother was determined following clinical manifestations of allergy, persisting for at least 24 months: allergy against respiratory and/or food allergens manifested by various individual combinations of symptoms (e.g., hay fever, conjunctivitis, eczema, bronchitis, asthma); monitoring by an allergist; positive skin prick tests or positive specific IgE; and anti-allergic treatment before pregnancy. No difference in pregnancy length was observed between allergic and healthy mothers. The study was approved by the Ethical Committee of the Institute for the Care of Mother and Child (Prague, Czech Republic) and carried out with signed written informed consent of the mothers.

Approximately 15 mL of cord blood was collected via umbilical vein punction into sterile heparinized tubes immediately after birth, as described previously [31]. The mononuclear cell fraction was obtained from whole cord blood by density gradient centrifugation (Histopaque-1077; Sigma-Aldrich, St. Louis, MO, USA) and Treg were magnetically isolated for DNA/RNA isolation and culture assays.

### 2.2. Flow Cytometry

Proportional characteristics of Treg were analyzed with flow cytometry using a TregFlowEx kit (cat. no. ED7417; Exbio pls., Vestec, Czech Republic). Following the manufacturer’s instructions, non-stimulated whole blood samples were stained for cell surface markers CD4 (labeled with fluorescein isothiocyanate—FITC; antibody clone MEM-241) and CD25 (labeled with phycoerythrin—PE; antibody clone MEM-181), fixated, permeabilized and stained for intracellular markers FOXP3 (labeled with allophycocyanin—APC; antibody clone 3G3) and Helios (cat. no. T7-771-T100; Exbio; labeled with phycoerythrin-cyanin 7—PE-Cy7; antibody clone 22F6).

Representative gating used for relative quantification of total Treg and Helios^+^/Helios^−^ subpopulations is shown in Appendix A. Firstly, the lymphocyte gate was set based on forward-scatter (FCS) and side-scatter (SSC) characteristics with doublets exclusion (FCS-A × FCS-H; Appendix A). Treg were gated from the lymphocyte gate as CD4^+^CD25^+^FOXP3^+^ cells (Appendix A), and median of fluorescent intensity (MFI) of FOXP3 in CD25^+^FOXP3^+^ Treg was determined (representative histograms for cord blood and maternal peripheral blood and fluorescence minus one (FMO) control are shown in Appendix A). Finally, Helios^+^ and Helios^−^ cells were gated and relatively quantified from the whole Treg population (Appendix A). The MFI of Helios was also determined in the total CD25^+^FOXP3^+^ Treg population. Representative histograms and FMO control are shown for samples of cord blood from neonates of healthy and allergic mothers (Appendix A); male and female newborns (Appendix A); and cord blood and maternal peripheral blood (Appendix A).

Flow cytometry data were acquired on a BD FACSCanto II flow cytometer using BD FACS Diva version 6.1.2 software (Becton Dickinson, Franklin Lakes, NJ, USA) and analyzed using FlowJo 7.2.2 (TreeStar, Ashland, OR, USA).

### 2.3. Magnetic Bead-Based Cell Isolation

Treg (CD4^+^CD25^+^CD127^low^) and target cells (CD4^+^CD25^−^ conventional T cells; Tconv) were magnetically isolated from cord blood mononuclear cells using an EasySep™ Human CD4^+^CD127^low^CD25^+^ Regulatory T Cell Isolation Kit (StemCell, Vancouver, BC, Canada). Obtained Treg were used for DNA/RNA extraction and for assessing Treg cytokine production and capacity for suppressing production of selected effector cytokines in a functional assay based on coculture of Treg with Tconv.

### 2.4. Genomic DNA and Total RNA Isolation from Treg

One million magnetically isolated Treg were used for genomic DNA extraction and total RNA extraction according to the manufacturer’s instructions using an AllPrep DNA/RNA Mini Kit (cat. no. 80204; Qiagen, Hilden, Germany). Total DNA concentration was estimated by spectrophotometric measurement at 260 nm, assuming that 50 μg of DNA per milliliter equals one absorbance unit. Purity of isolated DNA was estimated by the ratio of absorbance at 260 and 280 nm and was in the range of 1.8–2.3. Total RNA concentration was estimated by spectrophotometric measurement at 260 nm, assuming that 40 μg of RNA/ml equals one absorbance unit. RNA purity was likewise assessed by the ratio of absorbance at 260 and 280 nm and was in the range of 1.9–2.2. Isolated nucleic acids were stored in aliquots at −20 °C prior to downstream processing: reverse transcription followed by qPCR expression analysis of selected genes (RNA); or bisulphite modification followed by TSDR demethylation analysis using high-resolution melting (HRM) PCR (DNA).

### 2.5. Real-Time qPCR Gene Expression Analysis

Gene expression of cytokines in Treg isolated from cord blood was performed as described by Hrdy et al. [40]. Briefly, total RNA was reverse transcribed using a High Capacity RNA to cDNA kit (ThermoFisherScientific, Waltham, MA, USA). Gene expression of *Il10*, *Tgfb* and *Il35* (*p35* subunit) was determined using TaqMan probes (*Il10* Hs 00174086_m1, *Tgfb* Hs 00171257_m1 and *Il35* Hs 00168405_m1); actin beta Hs 99999903_m1 was used as an endogenous control (all ThermoFisherScientific). PCR reactions were run, and data were analyzed as described previously [31].

### 2.6. Bisulphite Conversion of Treg DNA

Bisulphite conversion of Treg genomic DNA was performed using an EpiTect Bisulfite Kit (48) (cat. no. 59104, Qiagen) according to the manufacturer’s instructions. Briefly, 0.5–1 μg DNA was mixed with 85 μL of bisulphite mix and 35 μL of DNA protect buffer and incubated in a thermal cycler programmed according to the manufacturer’s instructions. Converted DNA was stored at −20 °C before HRM-PCR analysis.

### 2.7. High-Resolution Melting Analysis of TSDR Demethylation

Methylation status of the TSDR of *FOXP3* in bisulphite-converted Treg genomic DNA was analyzed using an EpiTect HRM PCR Kit (100) (cat. no. 59445, Qiagen) according to the manufacturer’s instructions. Briefly, 10 ng of bisulphite-converted DNA was added to 5 μL of EpiTect HRM PCR Master Mix and a mixture of forward and reverse primers at a concentration of 0.75 μM each; the reaction mixture was topped to a total volume of 10 μL per reaction with RNase-free water. The real-time cycler was programmed as follows: 5 min incubation at 95 °C; followed by 45 cycles at 95 °C for 10 s, 55 °C for 30 s, 72 °C for 15 s; and, finally, high-resolution melting, performed from 65 to 95 °C, with a ramp rate of 0.02 °C/s and 25 acquisitions per second. All reactions were run in triplicates in a 384-well plate on LightCycler 480 (Roche, Basel, Switzerland).

The TSDR was considered as CNS2, located in *FOXP3* intron 1, as described by Baron et al. [11] and Wieczorek et al. [41] Primer pairs specific for the TSDR were designed to (1) not be methylation-specific themselves and (2) cover a sufficient number of potential methylation sites (i.e., 7 CpG sites). Primers were synthesized to order by KRD ltd (Prague, Czech Republic) with the following sequences: 5′-GGATGTTTTTGGGATATAGATTATGTTTTTAT-3′ (forward primer); and 5′-TATAAAATAAAATATCTACCCTCTTCTCTTCC-3′ (reverse primer), and used at a concentration of 0.75 μM each, as per the manufacturer’s instructions for HRM-PCR.

Methylated and demethylated bisulphite-converted human control DNA from the Epitect PCR Control DNA (cat. no.59695, Qiagen) set was mixed in varying ratios for use as a standard curve. Melting peaks specific for methylated products and demethylated products, differing in melting temperature, were detected and characterized using LightCycler^®^ 480 Software, version 1.5.0.39 (Roche). Proportion of demethylated TSDR (in %) was then calculated from comparison of area under the curve of the methylated and demethylated DNA-specific melting peaks.

### 2.8. Cell Culture

The target cells were cocultured with Treg as described previously [42]. Briefly, Tconv were plated into 24-well plates with or without Treg and cultivated for 72 h in RPMI medium (Sigma-Aldrich) supplemented with 10% fetal calf serum (Cambrex, East Rutherford, NJ, USA), gentamycin (Sigma-Aldrich, 40 mg/L) and L-glutamine (Sigma-Aldrich, 2 mM). An amount of 20 ng of recombinant human IL-2 (PeproTech, Rocky Hill, NJ, USA), 1 μg of purified, functional-grade human anti-CD3 (clone OKT3; ThermoFisherScientific) and 1 μg of purified, functional-grade human anti-CD28 (clone CD28.2; ThermoFisherScientific) were added per 10^6^ target cells to stimulate proliferation and cytokine production. After 72 h, culture supernatants were collected, and levels of selected cytokines were analyzed by ELISA.

### 2.9. ELISA

The cell culture supernatants were stored at −20 °C. Levels of IL-10, TGF-β, TNF-α, IL-1β and IL-4 in cell culture supernatants were quantified by ELISA, as described previously [43].

### 2.10. Statistics

Statistical and graphical analysis was performed using GraphPad Prism 6.0 software (GraphPad Software, La Jolla, CA, USA). Differences between groups were compared using the unpaired Student’s t-test or standard ANOVA in case of data with normal distribution (total Treg, iTreg and nTreg ratios, FOXP3 and Helios MFI, DNA demethylation analysis) and the nonparametric Kruskal–Wallis test for the rest of the data (culture supernatant levels of cytokines). Pearson’s correlation coefficient was calculated for all correlations. Results are presented as column plots showing mean + SEM for most of the data and as scatter plots in the case of all correlations.

## 3. Results

In our study, we compared immunological characteristics of Treg in the cord blood of newborns of allergic mothers (children with higher risk of allergy development, n = 108) with Treg in the cord blood of children of healthy mothers (children with lower risk of allergy development, n = 118). Chiefly, we used flow cytometry to analyze the total CD25^+^FOXP3^+^ Treg proportion and also evaluated the ratio of Helios^+^ Treg (putative nTreg) and Helios^−^ Treg (putative iTreg). The MFI (median of fluorescence intensity) of FOXP3 and Helios, transcription factors crucial for Treg phenotype stability and subpopulation homeostasis, were also compared between the groups. Furthermore, we compared the same immunological characteristics between male (n = 104) and female (n = 122) children. Later in the study, collection of maternal peripheral blood samples was added to the study setup, in order to assess the interdependence of parameters of maternal and neonatal immune systems in allergic (n = 85) as well as healthy (n = 82) mothers. To further evaluate the functional and developmental maturity of the neonatal immune system, we measured the effect of Treg on the secretion of selected cytokines in a Treg/Tconv cocultivation assay. Gene expression of IL-10, TGF-β and IL-35 (subunit p35) at the mRNA level was analyzed in magnetically isolated Treg by real-time qPCR. Finally, analysis of demethylation of the TSDR of the *FOXP3* promoter was performed in a smaller subset of samples.

In our study, nTreg and iTreg were identified as Helios^+^ and Helios^−^ subsets, adding up to 100% of CD25^+^FOXP3^+^ Treg. Therefore, nTreg, by definition, represent a complementary population to iTreg and reflect iTreg in all relevant characteristics. For the sake of clarity, and since induced Treg hold particular relevance in the context of allergy control, we elected to present graphs showing iTreg in the main body of the article and include the corresponding graphs of nTreg as Appendix A.

### 3.1. Treg Population Ratios in Cord Blood

No statistically significant difference in total CD25^+^FOXP3^+^ Treg (Figure 1a), Helios^−^ iTreg (Figure 1b) or Helios^+^ nTreg (Appendix A) was observed between the cord blood of children of allergic and healthy mothers. Although total Treg did not show any difference between male and female newborns (Figure 1c), iTreg were found to be significantly higher in female newborns (Figure 1d, *p* = 0.0098); correspondingly, male newborns had significantly higher nTreg (Appendix A, *p* = 0.0099). No difference was observed among the four groups for total Treg (Figure 1e). When divided according to sex as well as maternal allergy status, the significant differences between iTreg and nTreg were also present between male and female children of healthy (Figure 1f, *p* = 0.0121 and Appendix A, *p* = 0.0123, respectively) but not allergic mothers.

### 3.2. Treg Population Ratio Comparison between Cord Blood and Maternal Peripheral Blood

We saw no difference in total Treg, iTreg or nTreg (Appendix A, respectively) in the maternal peripheral blood of allergic and healthy mothers. Nevertheless, when cord blood Treg population characteristics were compared with those of the newborns’ mothers, a significantly lower proportion of total Treg (Figure 2a, *p* < 0.0001) as well as nTreg (Appendix A, *p* < 0.0001) was found in maternal peripheral blood, with iTreg (Figure 2b, *p* < 0.0001) being correspondingly higher in maternal samples. The highly significant differences between cord blood and maternal peripheral blood (*p* < 0.0001) were also preserved upon subdivision according to maternal allergy status (see Figure 2c,d and Appendix A for total Treg, iTreg and nTreg, respectively) as well as according to the sex of the newborns (see Figure 2e,f and Appendix A for total Treg, iTreg and nTreg, respectively).

### 3.3. Median of Fluorescence Intensity of FOXP3 and Helios

While we saw no statistically significant difference in the MFI of FOXP3 (Figure 3a) or Helios (Figure 3b) between Treg of newborns of allergic and healthy mothers, when children were divided according to sex, a significantly higher MFI of Helios (Figure 3d, *p* = 0.0297) but not FOXP3 (Figure 3c) was observed in male children. Furthermore, when comparing cord blood with maternal peripheral blood, maternal blood Treg had a significantly higher MFI of FOXP3 (Figure 3e, *p* = 0.0008) and a significantly lower MFI of Helios (Figure 3f, *p* < 0.0001) than cord blood Treg, although no difference was observed between peripheral blood of allergic and healthy mothers (Appendix A).

Analysis of the groups subdivided according to maternal allergy status and sex, shown in Appendix A, revealed neither a difference in the MFI of FOXP3 nor in the MFI of Helios when solely cord blood was considered (Appendix A, respectively). Comparison of cord blood and maternal peripheral blood, however, showed that the higher FOXP3 MFI in peripheral blood is more pronounced for allergic mothers (Appendix A) and mothers bearing female offspring (Appendix A). The higher cord blood Helios MFI seems independent of allergy status (Appendix A) but is more pronounced in male children than in female (Appendix A).

### 3.4. Correlation of Immune Parameters

To assess the interplay of the intrinsic and maternal factors influencing neonatal Treg populations in more depth, we performed correlations of the various parameters measured with flow cytometry, both in pooled data and in samples divided according to maternal allergy status and according to the newborns’ sex.

Since Helios^−^ induced Treg and Helios^+^ natural Treg are defined as complementary populations adding up to 100%, only correlations of iTreg with other parameters will be included in the main body of the article; the complementary correlations of nTreg are shown in Appendix A.

The correlation matrix including Pearson’s correlation coefficients and *p*-values of all correlation analyses performed is included in Appendix A.

### 3.5. Correlation of Immune Parameters in Cord Blood

No correlation was observed between Treg and iTreg (Appendix A), between Treg and nTreg (Appendix A) or between the FOXP3 MFI and iTreg (Appendix A) and the FOXP3 MFI and nTreg (Appendix A), regardless of newborns’ sex or maternal allergy status. A significant, although slight, positive correlation was found between the FOXP3 MFI and total Treg (Figure 4a, r = 0.1400, *p* = 0.0354); this correlation was not significant in children of allergic mothers but was significant and more pronounced in children of healthy mothers (Figure 4b, r = 0.2271, *p* = 0.0134), and an even stronger significant positive correlation was revealed between the two in male but not female children (Figure 4c, r = 0.2762, *p* = 0.0045). While there was no correlation between the Helios MFI and total Treg in any of the groups (Appendix A), unsurprisingly, a strongly significant negative correlation was found between the Helios MFI and iTreg when data were analyzed regardless of maternal allergy status and newborns’ sex (Figure 4d, r = −0.3833, *p* < 0.0001). This negative correlation was comparable when the data were analyzed according to maternal allergic status (Figure 4e, r = −0.3985, *p* < 0.0001 for healthy mothers, r = −0.3774, *p* < 0.0001 for allergic mothers) as well as according to the newborns’ sex (Figure 4f, r = −0.3888, *p* < 0.0001 for male children, r = −0.3814, *p* < 0.0001 for female children). Due to the role of Helios for nTreg characteristics, as well as its use in nTreg identification in our study, it is of little surprise that there is an equally strong and significant positive correlation between the Helios MFI and nTreg in the pooled samples (Appendix A) and in samples analyzed according to maternal allergic status (Appendix A) as well as according to the newborns’ sex (Appendix A).

For an overview of the correlation coefficients and *p*-values of all cord blood sample correlations, see Appendix A.

### 3.6. Correlation of Immune Parameters between Cord Blood and Maternal Blood

No correlation was seen between cord blood and maternal peripheral blood total Treg when data were not subdivided according to maternal allergy status and newborns’ sex (Appendix A). A slight but significant positive correlation can be seen between cord blood and maternal peripheral blood total Treg in healthy but not allergic mothers (Appendix A); no such correlation can be seen when children are divided according to sex (Appendix A). Induced Treg in cord blood and in maternal blood correlate positively, regardless of group subdivisions (Figure 5a, r = 0.2923, *p* = 0.0002 for undivided samples). The interdependence is somewhat stronger in healthy mothers (r = 0.3825, *p* = 0.0005) than in allergic mothers (r = 0.2249, *p* = 0.0423), both shown in Figure 5b. Lastly, the correlation between cord blood and maternal peripheral blood iTreg is only slightly stronger in female (Figure 5c, r = 0.2933, *p* = 0.0061) than in male newborns (Figure 5c, r = 0.2591, *p* = 0.0238). As nTreg, by definition, represent a complementary population to iTreg, it is not surprising that both correlation coefficients and measures of significance of the correlation of nTreg between cord blood and maternal peripheral blood are comparable with those of iTreg, regardless of whether they are considered in all samples (Appendix A), samples subdivided according to maternal allergy status (Appendix A) or samples subdivided according to the sex of the newborns (Appendix A).

For an overview of the correlation coefficients and *p*-values of all correlations between cord blood and maternal peripheral blood samples, see Appendix A.

### 3.7. Treg Functional Parameters: Coculture Assay and Gene Expression Analysis

Since Treg functional (i.e., suppressive) properties are more likely to prove informative regarding Treg maturation status and/or the risk for future allergy development, we performed an assay based on cocultivation of magnetically isolated Treg with magnetically isolated non-Treg CD4^+^ T cells (Tconv) at a 1:5 Treg/Tconv ratio. Selected relevant cytokines were detected by ELISA in the culture supernatants. Additionally, real-time qPCR analysis of gene expression of regulatory cytokines IL-10, TGF-β and IL-35 (p35 subunit) was performed in mRNA isolated from a magnetically separated population of Treg (CD4^+^CD25^+^CD127^low^).

Although there was no significant difference in IL-10 concentration between the samples with Treg and the samples including only stimulated Tconv, regardless of maternal allergy status, an observable, though non-significant, trend toward lower concentration of this chief regulatory cytokine was visible between samples isolated from the cord blood of children of allergic mothers compared to children of healthy mothers (Figure 6a). Regrettably, no difference was detected in TGF-β concentration among any groups (Figure 6b). Among the inflammatory and Th2 cytokines selected, we observed a significant increase in TNF-α in samples with only stimulated Tconv but not Tconv with Treg (Figure 6c). Slightly surprisingly, this was observed in samples isolated from the cord blood of children of healthy but not allergic mothers, although there was a similar trend in this group. IL-1β (Appendix A) and IL-4 (Appendix A), chosen as additional inflammatory and Th2 cytokines, respectively, showed no statistically significant difference.

Furthermore, gene expression of *Il10*, *Tgfb* and *Il35* (*p35* subunit) (Appendix A, respectively) was compared. Although the difference was not significant, a slight trend towards lower expression of regulatory cytokines in Treg isolated from the cord blood of children of allergic mothers is discernible.

### 3.8. Epigenetic Analysis of TSDR Demethylation

Considering the importance of epigenetic control of Treg maturation, stability, function and homeostasis, particularly demethylation of the TSDR of the FOXP3 promoter, we performed epigenetic analysis of this locus. Although we found no difference in TSDR demethylation in DNA isolated from cord blood Treg, regardless of maternal allergy status (Figure 7a) and the child’s sex (Figure 7b), as well as their combination (Appendix A), we observed a significant correlation between TSDR demethylation and total Treg (Figure 7c, r = 0.4165, *p* = 0.0428), supporting the role of TSDR demethylation in Treg maturation already in the perinatal period.

## 4. Discussion

Regulatory T cells play a key role in inducing and maintaining immune tolerance toward innocuous stimuli, including allergens [5]. While it has been shown that the critical role of Treg can be traced to prenatal and perinatal periods [37,44,45,46], the details of the complex processes accompanying the perinatal maturation of Treg and their subpopulations are yet to be elucidated in detail.

Contrary to our previous observations [42,47], we did not see any difference in Treg, iTreg or nTreg proportions between the cord blood of children of healthy and allergic mothers. This discrepancy might be attributed to several factors. Importantly, in previous studies, mothers delivering vaginally were included, whereas in the current study, mothers gave birth via cesarean section. While the effect of cesarean section on the proportional characteristics and function of T cell subpopulations has yet to be elucidated and studies regarding the effect on Treg in particular are limited (e.g., [48,49]), there is a profound difference between cesarean section versus vaginal delivery regarding their effect on immune parameters, in the short term, likely due to stress-related and hormonal factors [50]. Importantly, cesarean section has been reported to be associated with an increased risk of immune-related diseases in children, including asthma [51] and allergic rhinitis [52]. Increased risk of allergy and differential immune regulation stemming from cesarean section may thus mask the differences between Treg population proportions in the cord blood of children of allergic and healthy mothers, which we observed in previous studies performed on newborns born vaginally.

Other potential sources of variability include differences in gating and the method of staining utilized, factors known to strongly influence the variability among the published studies [45,47,53,54,55,56]. Notably, due to technical limitations of the staining panels used in this study, we were unable to use the CD127 cell surface marker for Treg identification. Further confounding factors, affecting the comparability of studies performed by various groups and in various settings, include the clones of the anti-FOXP3 antibodies used [57,58] or environmental factors affecting the mothers during pregnancy (e.g., difference between studies performed in rural areas [54] or on urban cohorts [53]). Considering this heterogeneity and generally higher interindividual variability inherent to human probands as opposed to experimental animal models, there is a pressing need for the introduction of a unified, widely accepted methodical framework for human Treg identification and study by flow cytometry as well as rigorous data analysis.

In our study performed mostly on an urban cohort of probands, we observed a slight but significant correlation between the MFI of FOXP3, the chief Treg lineage-specific factor, and the total Treg proportion in cord blood. This correlation was stronger in the cord blood of children of healthy mothers and particularly in male children but was not significant in children of allergic mothers and female children. In addition, we found a slightly but significantly lower proportion of iTreg, as well as a higher proportion of nTreg, in male children regardless of the allergy status of their mothers. This was also apparent when analyzing the MFI of transcription factor Helios: male newborns exhibited significantly higher values of the Helios MFI in cord blood Treg than female children, while no difference was seen between children of healthy and allergic mothers. The Helios MFI, in fact, showed a consistent, significant positive correlation with the nTreg population in cord blood and, correspondingly, a strong negative correlation with iTreg.

Importantly, we observed a significantly larger proportion of iTreg and a lower proportion of nTreg in maternal peripheral blood compared with cord blood, as well as higher values of the Helios MFI in cord blood than in maternal blood. While the peripheral blood of mothers at birth is itself an immunologically highly specific kind of adult blood sample [59], it undeniably exhibits a much more mature regulatory state than cord blood. Therefore, these findings corroborate our previously stated hypothesis that a higher proportion of Helios^+^ nTreg and a corresponding higher positivity of Helios reflect a less mature state of immune regulation [42,47]. This putative immaturity, more pronounced in male than female children in our study, reveals itself in decreased iTreg, as these cells are mostly induced when mature but antigenically naïve T cells encounter harmless antigens under appropriate tolerogenic circumstances in peripheral tissues [18,60]. Complementarily, male newborns had a larger nTreg population with a higher Helios MFI.

Interestingly, a higher incidence of various types of allergic disorders has been reported in male children. Studies including meta-analyses have described this phenomenon in bronchial asthma [61,62,63,64] and allergic rhinitis [65,66], where the higher incidence in boys generally persists until the onset of puberty. Furthermore, boys were reported to have a higher incidence of certain types of food allergy [67,68,69,70], atopic dermatitis [71] and even anaphylaxis in a group aged 0–9 years [72]. A higher incidence of allergies in male children has been observed in cohorts differing significantly in their genetic and environmental background. It can thus be surmised that the underlying mechanisms are intrinsic rather than environmental and quite robust. We propose that a lower maturation status of immune regulation, reflected by lower iTreg, higher nTreg and a higher Helios MFI, may be one of the factors behind the increased risk of sensitization and early-age allergy development in male children. Consequently, the higher significance of the correlation between the FOXP3 MFI and total Treg among children of healthy mothers, and particularly male children with higher risk of developing certain kinds of allergies at an early age [66,67,70,73,74], may be due to a drive for Treg maturation and induction in an effort to compensate for their subpar functional maturity.

Although the magnitude of the difference between nTreg in male and female children is not very high, to the best of our knowledge, our findings present the first report of an increased Helios^+^ nTreg Helios MFI in cord blood Treg of male newborns, and, indeed, it is the first analysis of sex-dependent differences in Helios-expressing cells in cord blood. More extensive, rigorously conducted confirmatory studies will be needed to evaluate the clinical relevance of this effect, and inclusion of further surface markers such as CD127, CD45RA, CTLA-4, PD-1 and LAG-3 would provide better population identification and more insight into the activation status and biological significance of these cells. It would also be of interest to compare cord blood samples of newborns delivered vaginally with the current cohort of probands who were delivered with cesarean section.

Interestingly, while maternal peripheral blood contained significantly fewer Treg than cord blood, the MFI of FOXP3 was higher in maternal blood. Sufficient FOXP3 expression is key for Treg functional efficiency and phenotypic stability [21,75,76,77]. Therefore, the fully developed, adult steady-state immune system may need fewer Treg due to their more stable and fully pronounced regulatory properties. Alternatively, the lower proportion of total Treg in mothers might be due to pre-birth (third trimester) changes in maternal immune regulation, which include Treg subtraction in preparation for the induction of birth [78,79], a highly controlled inflammatory process [78,80].

We were also able to uncover correlations between the Treg population and iTreg as well as nTreg subpopulations in cord blood and maternal blood, consistent with the well-known tight inter-regulation of the maternal and fetal immune systems [81,82]. In particular, we observed a positive correlation between the total Treg population proportion in the cord blood and maternal peripheral blood of healthy but not allergic mothers. Furthermore, we saw a significant positive correlation between iTreg and nTreg populations in cord blood and maternal peripheral blood; these correlations were stronger in healthy as opposed to allergic mother–child pairs and only slightly stronger in the case of female newborns. Considered together, these findings support strong interdependence of maternal and fetal immune regulation which persists into the perinatal period. The somewhat stronger correlation seen in the case of children of healthy mothers and also female children might represent a minute, yet important proof of a tighter interdependence seen in these neonatal groups, conceivably contributing to their higher maturation and lower risk of later allergy development.

FOXP3 is generally accepted as the chief Treg lineage-specific transcription factor. Nonetheless, it has been shown that FOXP3 expression can transiently be upregulated in activated T cells [83,84]. As the same has been described for CD25 [83], the higher proportion of CD25^+^FOXP3^+^ Treg described in cord blood may include a population of activated non-Treg CD4^+^ T cells in the neonatal circulation rather than bona fide Treg. Since epigenetic regulation (particularly TSDR demethylation) is a more reliable marker of stable, lineage-committed Treg than simple FOXP3 positivity [11,85,86], TSDR demethylation analysis of a sufficient number of samples would be necessary to conclusively distinguish the two possibilities. Similarly, while Helios has originally been described as a fairly specific marker of human natural Treg [20,87], its usefulness for distinguishing nTreg and iTreg has been questioned by other studies which claimed it to be another marker of recent T cell activation, transiently upregulated in CD4^+^ T cells [88,89]. Higher amounts of Helios^+^FOXP3^+^ T cells could then, in fact, signify a higher number of activated, potentially inflammatory T cells in the higher-risk groups. To this day, the real biological significance of Helios has not been conclusively determined [90], though our data seem to support its role as an nTreg-specific transcription factor, in line with recent studies describing differences in TCR specificity of Helios^+^ and Helios^−^ cells consistent with the functional role of nTreg and iTreg, respectively [91,92].

While comparison of proportional and phenotypic characteristics of Treg and Treg subpopulations can provide valuable insight into neonatal homeostasis, it is increasingly clear that the functional capacity of these cells is more likely to hold predictive value for later allergy development. This can be assessed both indirectly (e.g., by measuring the intracellular presence of key regulatory cytokines IL-10, TGF-β and IL-35 [93] in Treg by flow cytometry; detecting secretion of these cytokines in plasma or culture supernatants; analyzing gene expression of regulatory cytokines with real-time qPCR; evaluating expression of functionally relevant Treg cell surface molecules including CTLA-4 and PD-1 [94,95,96], GITR, [97] LAG-3 [98,99], CD39 and CD73 [8] by flow cytometry) or directly through assays based on coculture of Treg with target cells and measuring the suppression of effector functions (e.g., proliferation, effector cytokine secretion). In our coculture-based assay, we did not see any significant difference between stimulated conventional T cells cultured with or without Treg for any studied parameters, although a nonsignificant trend toward lower IL-10 secretion was discernible in samples isolated from the cord blood of children of allergic mothers, consistent with our hypothesis of more robust regulatory cells in the low-risk group. We were likewise unable to confirm any Treg-mediated suppressive effect toward effector cytokines TNF-α, IL-1β or IL-4, although there was an observable, though non-significant, trend towards lower TNF-α production when stimulated target cells were cocultured with Treg.

Furthermore, expression of *Il10*, *Tgfb* and *Il35* genes was discernibly, albeit non-significantly, lower in Treg isolated from the cord blood of children of allergic mothers. IL-10, TGF-β and IL-35 [93] represent major regulatory cytokines and are all secreted by Treg as well as other regulatory cell populations [100,101,102,103]. IL-10 release is particularly important for in vivo suppressive properties of Treg, and IL-10 production by a subset of Treg (so-called Tr1 cells) or regulatory B cells plays a crucial role in the clinical success of allergen-specific immunotherapy [101,104,105,106], the sole causal therapeutic approach to allergic disorders. TGF-β plays a key role in iTreg induction on the periphery [18] and general maintenance of peripheral, particularly mucosal, tolerance [60,107]. IL-35 is a recently described regulatory cytokine belonging to the IL-12 interleukin family [108]. It has been implicated in the control of allergic rhinitis [109] and childhood asthma [108,110] and is currently being evaluated in the context of sublingual allergen immunotherapy [111]. The lower gene expression of these regulatory cytokines in the high-risk group further supports the notion of a lower functional capacity of their Treg.

Beside Treg function analysis, epigenetic control is currently extensively studied as the key determinant of Treg stability and function [9,10,26,28,85]. In particular, the demethylation status of conserved noncoding sequences in the Treg-specific demethylated region (TSDR) has been proposed as a key marker of Treg lineage stability and functional efficacy [11,85,86,112], possibly capable of discerning bona fide, lineage-determined CD25^+^FOXP3^+^ Treg from recently activated effector T cells, in which CD25, FOXP3 and even Helios might be transiently upregulated [11,83,88,113,114].

In our study, we analyzed TSDR demethylation in genomic DNA isolated from Treg isolated from the cord blood of a subset of samples. Although we found no difference in TSDR demethylation between either children of healthy and allergic mothers or male and female children, we noted a significant positive correlation between the percentage of TSDR demethylation and proportion of total Treg in the cord blood, confirming that TSDR demethylation influences Treg population homeostasis as early as in the perinatal period. Furthermore, our data hinted at a non-significant, but discernible, positive correlation between TSDR demethylation and the cord blood nTreg population proportion, supporting the published hypothesis that TSDR demethylation plays a greater role in the homeostasis and lineage stability of nTreg compared with iTreg [23,24,25].

Taken together, our data further support our working hypothesis that cord blood Treg and, by extension, the entire neonatal immune regulation are immature, and that this may contribute to the risk of allergy development. We did not confirm this in the case of children of allergic mothers, suggesting that clinical setting (i.e., cesarean section as opposed to vaginal delivery) as well as methodical differences in Treg identification and analysis may underlie the significant variability observed among the many published studies. Our data instead suggest a slightly decreased maturation status in male children compared to females, a finding consistent with the reported higher risk of male infants to develop various kinds of early-age allergic diseases [63,64,65,66,67,68,69,70,71]. More studies will nonetheless be needed to fully understand the exact significance of Treg in this phenomenon. The novel observation of slightly but detectably lower Helios^−^ Treg population in the cord blood of male newborns is of special interest. Further research into sex-dependent differences in regulatory T cell characteristics in cord blood is warranted.

The observed correlation of iTreg and nTreg subpopulations between cord blood and maternal peripheral blood supports the generally accepted notion of a tight interconnection of maternal and fetal immune system regulation, which, under physiological conditions, may prime neonatal immunity to react to harmless exogenous antigens in a tolerogenic fashion or, in the case of maternal allergic inflammation, may contribute to the higher risk of allergy development in the children [81,82,115,116].

We acknowledge several limitations of our study. The discrepancy between Treg proportional and functional characteristics published in this study and findings reported in our previous studies [42,47] as well as studies by other groups [53,54] attests to the importance of methodical, as well as population- and biological context-dependent, sources of variability. Importantly, the functional and epigenetic assays reported in our study are limited by the relatively low number of individuals included. Further studies with higher numbers of experimental subjects are thus needed to conclusively evaluate the functional and epigenetic differences between cord blood Treg of children with higher risk of allergy development and children with lower risk of allergy development. Furthermore, while maternal allergic status is more important due to the intimate relationship between the mother and the child both prenatally and postnatally (e.g., due to breastfeeding), recent studies have demonstrated that a biparental positive allergy status may prove an even stronger predictor of an increased risk of allergy development [39]. Future studies might therefore benefit from including paternal and biparental allergy status rather than just maternal. Finally, long-term prospective studies capable of longitudinally following the immunological and allergologically relevant characteristics of the probands would be highly useful in determining the actual relevance of the descriptive, functional and epigenetic parameters measured at the moment of birth for prediction of allergy development during postnatal life.

## Figures and Tables

**Figure 1 biomedicines-09-00170-f001:**
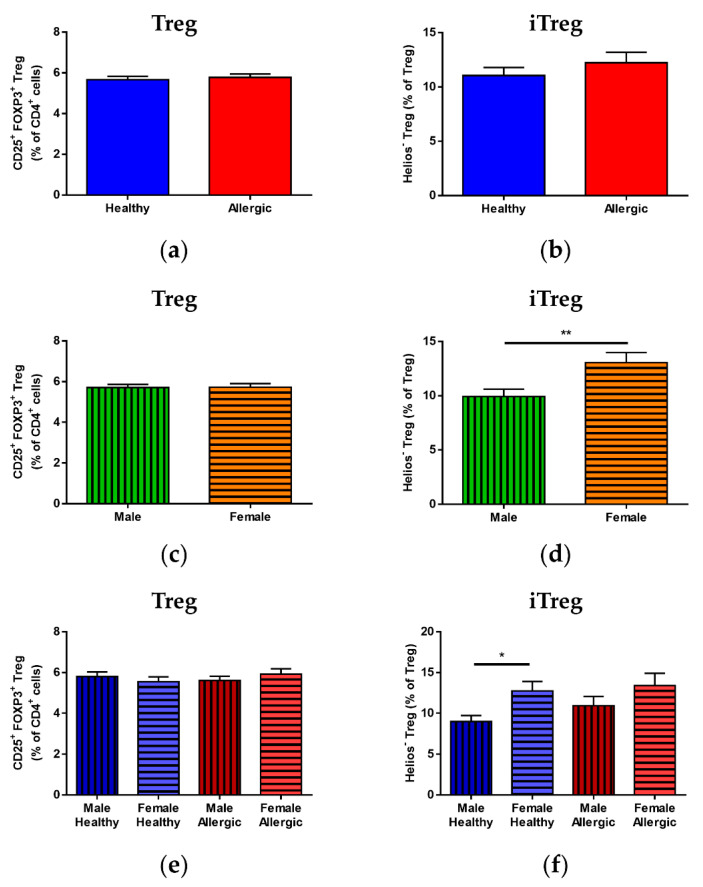
Proportions of total regulatory T cells (Treg) population and induced Treg (iTreg) and natural Treg (nTreg) subpopulations in cord blood of male and female children of healthy and allergic mothers. Samples of cord blood (n = 226) were stained and analyzed by flow cytometry. (**a**,**b**) Flow cytometry analysis of cord blood samples of children of healthy (n = 118) and allergic (n = 108) mothers. (**a**) Proportion of CD25^+^FOXP3^+^ Treg in the cord blood CD4^+^ T cell population. (**b**) Proportion of Helios^−^ iTreg in the cord blood Treg population. (**c**,**d**) Flow cytometry analysis of cord blood samples of male (n = 104) and female (n = 122) newborns. (**c**) Proportion of CD25^+^FOXP3^+^ Treg in the cord blood CD4^+^ T cell population. (**d**) Proportion of Helios^−^ iTreg in the cord blood Treg population. ** *p* = 0.0098. (**e**,**f**) Flow cytometry analysis of cord blood samples of newborns divided according to sex and maternal allergy status: male children of healthy mothers (n = 53), female children of healthy mothers (n = 65), male children of allergic mothers (n = 51) and female children of allergic mothers (n = 57). (**e**) Proportion of CD25^+^FOXP3^+^ Treg in the cord blood CD4^+^ T cell population. (**f**) Proportion of Helios^−^ iTreg in the cord blood Treg population. * *p* = 0.0121.

**Figure 2 biomedicines-09-00170-f002:**
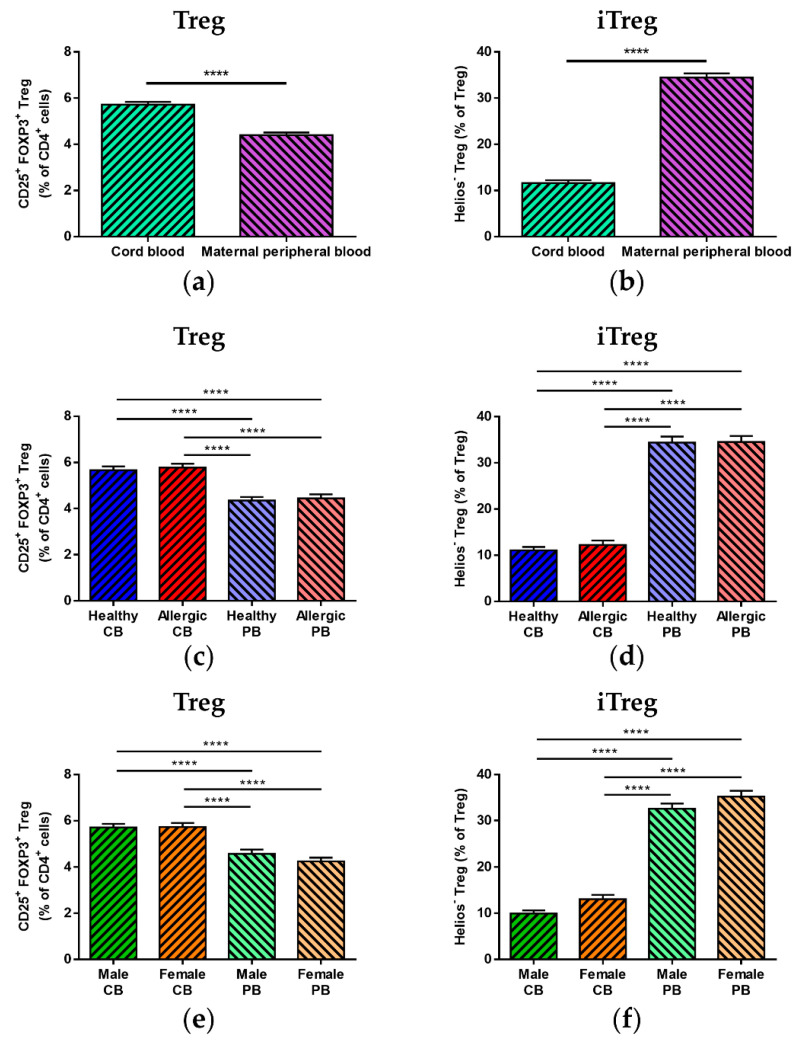
Comparison of total regulatory T cell (Treg) population and induced Treg (iTreg) and natural Treg (nTreg) subpopulations between cord blood and peripheral blood of healthy and allergic mothers. Samples of cord blood (n = 226) and maternal peripheral blood (n = 167) were stained and analyzed by flow cytometry. (**a**,**b**) Flow cytometry analysis of cord blood and maternal peripheral blood samples. (**a**) Proportion of CD25^+^FOXP3^+^ Treg in the CD4^+^ T cell population. **** *p* < 0.0001. (**b**) Proportion of Helios^−^ iTreg in total Treg population. **** *p* < 0.0001. (**c**,**d**) Flow cytometry analysis and comparison of samples of cord blood of children of healthy (n = 118) and allergic (n = 108) mothers and maternal peripheral blood of healthy (n = 82) and allergic (n = 85) mothers. (**c**) Proportion of CD25^+^FOXP3^+^ Treg in the CD4^+^ T cell population. **** *p* < 0.0001. (**d**) Proportion of Helios^−^ iTreg in total Treg population. **** *p* < 0.0001. (**e**,**f**) Flow cytometry analysis of cord blood and maternal peripheral blood samples divided according to the newborns’ sex: cord blood of male children (n = 104), cord blood of female children (n = 122), peripheral blood of mothers of male children (n = 76) and peripheral blood of mothers of female children (n = 86). (**e**) Proportion of CD25^+^FOXP3^+^ Treg in the CD4^+^ T cell population. **** *p* < 0.0001. (**f**) Proportion of Helios^−^ iTreg in total Treg population. **** *p* < 0.0001.

**Figure 3 biomedicines-09-00170-f003:**
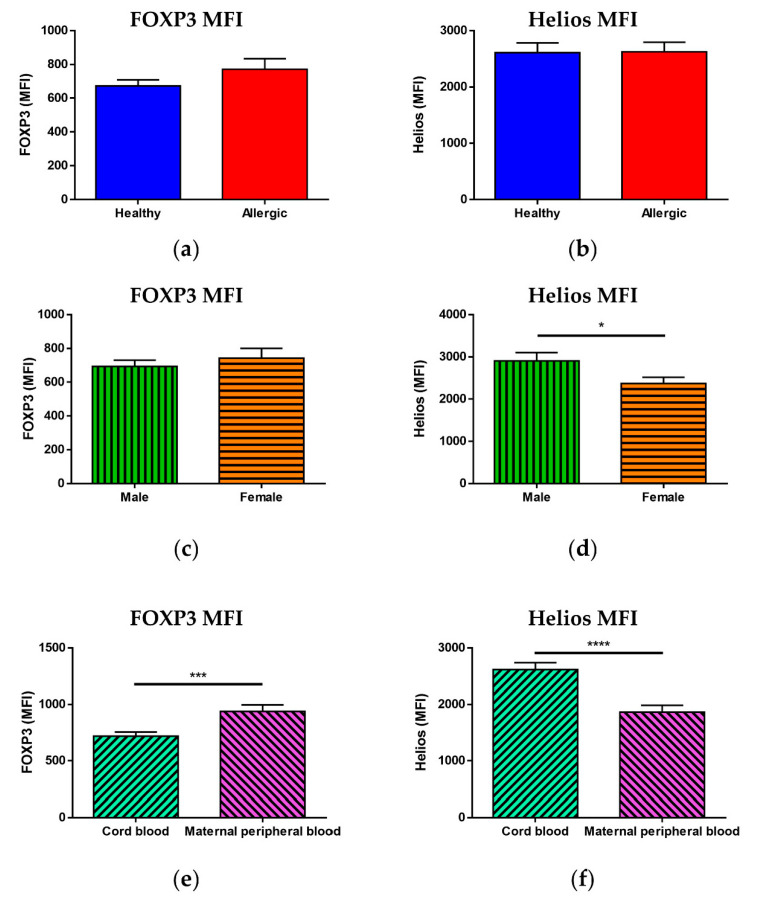
Median of fluorescence intensity of transcription factors FOXP3 and Helios in regulatory T cells (Treg)**.** Samples of cord blood (n = 226) and maternal peripheral blood (n = 167) were stained and analyzed by flow cytometry. (**a**,**b**) Flow cytometry analysis of cord blood samples of children of healthy (n = 118) and allergic (n = 108) mothers. (**a**) Median of fluorescence intensity (MFI) of FOXP3 in CD25^+^FOXP3^+^ Treg. (**b**) Median of fluorescence intensity (MFI) of Helios in CD25^+^FOXP3^+^ Treg. (**c**,**d**) Flow cytometry analysis of cord blood samples of male (n = 104) and female (n = 122) newborns. (**c**) Median of fluorescence intensity (MFI) of FOXP3 in CD25^+^FOXP3^+^ Treg. (**d**) Median of fluorescence intensity (MFI) of Helios in CD25^+^FOXP3^+^ Treg. * *p* = 0.0297. (**e**,**f**) Flow cytometry analysis of cord blood and maternal peripheral blood samples. (**e**) Median of fluorescence intensity (MFI) of FOXP3 in CD25^+^FOXP3^+^ Treg. *** *p* = 0.0008. (**f**) Median of fluorescence intensity (MFI) of Helios in CD25^+^FOXP3^+^ Treg. **** *p* < 0.0001.

**Figure 4 biomedicines-09-00170-f004:**
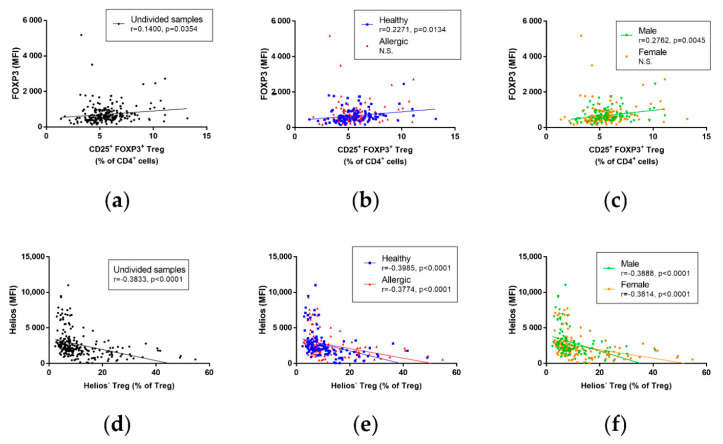
Correlation of selected regulatory immune parameters in cord blood of children. Characteristics of Treg cells in cord blood of male (n = 104) and female (n = 122) children of healthy (n = 118) and allergic (n = 108) mothers were measured using flow cytometry and correlated, utilizing Pearson’s correlation coefficient. (**a**) Correlation between CD25^+^FOXP3^+^ Treg and the MFI of FOXP3 in cord blood of children. r = 0.1400, *p* = 0.0354. (**b**) Correlation between CD25^+^FOXP3^+^ Treg and the MFI of FOXP3 in cord blood of children of healthy and allergic mothers. r = 0.2271, *p* = 0.0134 in children of healthy mothers. (**c**) Correlation between CD25^+^FOXP3^+^ Treg and the MFI of FOXP3 in cord blood of male and female children. r = 0.2762, *p* = 0.0045 in male children. (**d**) Correlation between Helios^−^ iTreg and the MFI of Helios in cord blood of children. r = −0.3833, *p* < 0.0001. (**e**) Correlation between Helios^−^ iTreg and the MFI of Helios in cord blood of children of healthy and allergic mothers. r = −0.3985, *p* < 0.0001 for healthy mothers, r = −0.3774, *p* < 0.0001 for allergic mothers. (**f**) Correlation between Helios^−^ iTreg and the MFI of Helios in cord blood of male and female children. r = −0.3888, *p* < 0.0001 for male, r = −0.3814, *p* < 0.0001 for female.

**Figure 5 biomedicines-09-00170-f005:**
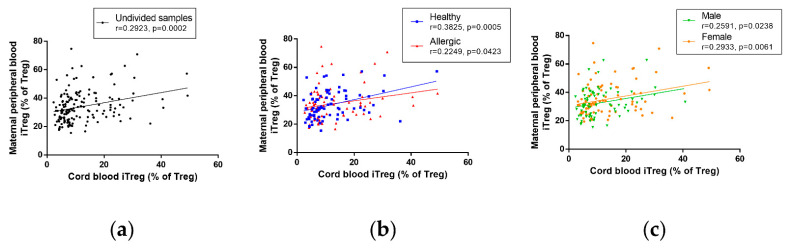
Correlation of induced regulatory T cells (iTreg) between cord blood of children and maternal peripheral blood of their mothers. Characteristics of Treg cells in cord blood of male (n = 76) and female (n = 86) children of healthy (n = 80) and allergic (n = 82) mothers were measured using flow cytometry and correlated with characteristics in maternal peripheral blood, utilizing Pearson’s correlation coefficient. (**a**) Correlation between Helios^−^ iTreg in cord blood of children and peripheral blood of their mothers. r = 0.2923, *p* = 0.0002. (**b**) Correlation between Helios^−^ iTreg in cord blood of children and peripheral blood of healthy and allergic mothers. r = 0.3825, *p* = 0.0005 for healthy mothers, r = 0.2249, *p* = 0.0423 for allergic mothers. (**c**) Correlation between Helios^−^ iTreg in cord blood of male and female children and peripheral blood of their mothers. r = 0.2591, *p* = 0.0238 for male children, r = 0.2933, *p* = 0.0061 for female children.

**Figure 6 biomedicines-09-00170-f006:**
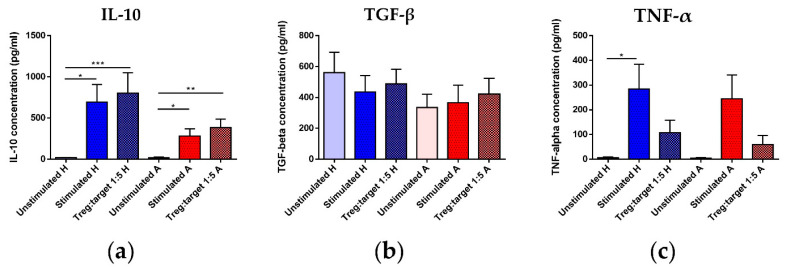
Concentration of selected cytokines in supernatants of regulatory T cells (Treg)/conventional T cells (Tconv) cocultures. Non-Treg CD4^+^ T cells (Tconv) magnetically isolated from children of healthy (n = 14) and allergic (n = 15) mothers were cultured with IL-2, unstimulated (negative control) or stimulated with a cocktail of functional-grade anti-CD3 + anti-CD28 antibodies (positive control). Treg magnetically isolated from the same cord blood samples were added to stimulated Tconv at 1:5 Treg/Tconv ratio. Culture supernatants were collected after 72 h of culture and concentrations of selected cytokines in the supernatant were measured with ELISA. (**a**) Concentrations of IL-10 in the supernatant of unstimulated Tconv, stimulated Tconv and Treg/Tconv cocultures of Tconv and Treg of children of healthy and allergic mothers. * *p* < 0.05, ** *p* < 0.01, *** *p* < 0.001. (**b**) Concentrations of TGF-β in the supernatant of unstimulated Tconv, stimulated Tconv and Treg/Tconv cocultures of Tconv and Treg of children of healthy and allergic mothers. (**c**) Concentrations of TNF-α in the supernatant of unstimulated Tconv, stimulated Tconv and Treg/Tconv cocultures of Tconv and Treg of children of healthy and allergic mothers. * *p* < 0.05.

**Figure 7 biomedicines-09-00170-f007:**
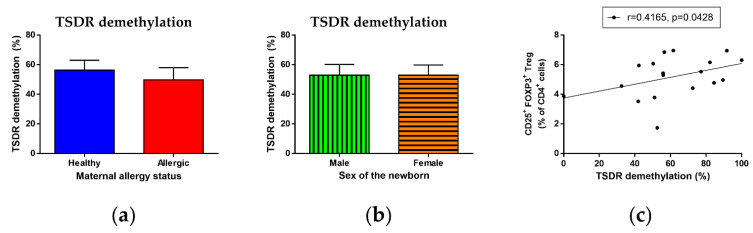
Epigenetic analysis of demethylation of Treg-specific demethylated region (TSDR) locus of FOXP3 promoter. Genomic DNA was obtained from Treg magnetically isolated from cord blood samples (n = 27). Bisulphite conversion of DNA was performed and methylation status of converted DNA samples was measured using high-resolution melting PCR analysis. (**a**) Percentage of TSDR demethylation of DNA isolated from children of healthy (13) and allergic (14) mothers. (**b**) Percentage of TSDR demethylation of DNA isolated from male (n = 18) and female (n = 9) children. (**c**) Correlation between percentage of TSDR demethylation and proportion of CD25^+^FOXP3^+^ Treg of CD4^+^ T cells in cord blood. r = 0.4165, *p* = 0.0428.

## Data Availability

The data presented in this study are available on request from the corresponding author due to privacy restrictions (clinical material from human probands).

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
