# Peer review of "Lower Functional and Proportional Characteristics of Cord Blood Treg of Male Newborns Compared with Female Newborns"

_biomedicines, 2021, doi:10.3390/biomedicines9020170_

Round 1

Reviewer 1 Report

The manuscript by Cerny et al. reports a study on Treg quantification and functional characterization in the cord blood of newborns divided on the basis of sex and of the allergic status of the mother. Comparison between Treg subpopulations in CB and mother peripheral blood was also performed. An attempt of functional characterization was performed by testing cytokine production by stimulated Tconv in the presence or absence of purified Treg (CD4+CD25hi). Finally, correlation analysis was performed between several parameters (i.e. Total Treg in CD4 T cells and Helios MFI), and analysis of TSDR demethylation in purified Tregs from a limited number of CB samples was performed. The phenotypic analysis was carried out on a high number of samples (>100 per group)

The main finding of the paper is that in cord blood from male newborns there was a lower percentage of iTreg and a minimal but significant increase of nTreg. Tregs (CD4+CD25+) were classified in nTreg and iTreg on the basis of Helios expression (Helios + nTreg and Helios – iTreg).

The other finding is that the cord blood % of Tregs was higher than in maternal peripheral blood and that maternal PB Treg contained a higher percentage of iTreg (Helios-) and a lower % of Helios + nTreg as compared to their CB counterpart. Consistent with these findings Helios MFI in Treg was higher in maternal PB compared to CB.  TSDR demethylation was not different CB Treg when comparing male versus female newborns but a positive correlation was found between demethylation of TSDR locus in DNA isolated from Treg and percentage of Treg in the CD4+ population.

The manuscript provides some interesting information, mainly regarding the difference in Treg subpopulation in male and female newborns CB but it has several limitations.

  1. 1. The difference that represents the main (only) finding of the study is minimal, and flow cytometry data are not shown except for a representative group of dot plots showing the gating strategy as a supplemental figure.  An example of the difference in MFI (Helios) between groups should be shown by a histogram plot.
  2. Another relevant concern regards the limitation of the functional analysis. Cytokine secretion in T conv stimulated with anti-CD3 /CD28 is detected only in CB from children from healthy mother (only IL-10 and TNF) and no suppression or significant variation is observed upon addition of Tregs.
  3. The evidence of a different percentage of iTreg and nTreg in male vs female newborns is novel but is not really discussed. No previous studies supporting this evidence are included in the reference list and the references cited in support to an increased male tendency to develop allergies at an early age are few and limited in their impact.
  4. There are differences with previous findings mentioned in the discussion that the authors ascribe to differences in technical protocols and methodological approaches. This is a major limitation. Therefore it is necessary to support the new evidence (male vs female Treg) with very clear data accurately describing and showing data analysis.

Minor points

-Figures are difficult to follow and should be reorganized for a clearer data representation.

-Many correlation analyses are reported, some of them not really adding new information. This part could be substituted with a table clearly reporting r and p values for each correlation. (As an alternative, interpolating lines together with the value of r and p for each correlation should be included in individual plots. Only the most relevant findings should be shown).

-Tregs are defined as CD25hi FOXp3+ but the only dot plot reported in supplemental data shows mainly CD25+ FOXp3+ CD4+ T cells. Please clarify.

Author Response

Firstly, we sincerely thank the editor and the reviewers for the time taken to read our manuscript and for formulating the comments, suggestions and critiques. We appreciate the positive comments and offered suggestions. We endeavoured to revise the manuscript to address the extensive and insightful concerns of the reviewers.

Point-by-point listing of all revisions made to the manuscript:

All changes are listed by appearance from the beginning to the end of the text. Lines containing the major changes to the revised manuscript are shown in bold for better orientation. Lines provided correspond to the numbering in the document showing all changes (i.e. the tracked copy of the revised manuscript).

  • Line 47: As a response to a comment of Reviewer 1, we substituted all instances of CD25highFOXP3+ in our text with the more correct CD25+FOXP3+. First example of this substitution can be found on the line number 47; the substitutions were consistently made in the body of the entire article as well as in figure captions and in the Supplementary files. The substitutions number 23 in the main article and 13 in the Supplementary files.
  • Lines 141-150: In response to a comment of Reviewer 1, we expanded our gating strategy presented in Supplementary figure 1 and added representative histograms of FOXP3 and Helios fluorescence intensity, showing differences between the various groups which we report in the main body of the article. The description of the gating strategy has been correspondingly expanded, as seen on lines 141-150. Captions of various panels were likewise changed to reflect the definitive panels included in Supplementary figure 1. Correspondingly, figure caption of Supplementary figure 1 was updated in the Supplementary files.
  • Lines 161-164: Deleted to maintain constant spacing between the paragraphs in the article.
  • Lines 165-188: To further support our functional assays, data regarding gene expression of Il10, Tgfb and Il35 (p35 subunit) on mRNA level was added to the supplementary files (Supplementary figure 11). Description of methodical aspects analysis was added to the Materials and Methods section of main body of the article, lines 165 through 188.
  • Line 224: FTS was substituted with the more correct foetal calf serum.
  • Line 228: State and address of the provider were removed since they are now listed upon the first occurrence on the added line 184.
  • Line 234: IL-1 was changed to more precise IL-1β.
  • Lines 260-261: A sentence describing the added gene expression analysis of regulatory cytokines was added.
  • Lines 264-269: Added to explain the changes made to figures 1 and 2 in response to the comment of Reviewer 1, which suggests that the figures should be reorganized and made more comprehensive for the reader. To comply, we moved the panels showing Helios+ nTreg population characteristics into newly included Supplementary figures 2 and 4. This change in graphical data presentation and inclusion of panels in the main body of the article and in the Supplementary files resulted in the necessity of re-labelling all subsequent references to Supplementary figure numbers, as well as references to the panels in Figures 1 and 2 in the main body of the article, starting on line 273.
  • Lines 273 through 317: References to individual figures and panels were re-labelled according to the reorganization of figures, as mentioned previously.
  • Line 299: Since the abbreviation MFI was introduced earlier, explanation of the abbreviation was removed for the sake of consistency.
  • Line 318: Panels showing nTreg parameters in Figure 1 were moved into Supplementary figure 2 for the sake of clearer data presentation. Correspondingly, figure caption of Supplementary figure 2 was updated in the Supplementary files.
  • Lines 320-335: Figure caption was updated to reflect the removal of panels showing nTreg. All relevant references to Supplementary figures were re-numbered. All instances of CD25highFOXP3+ were substituted with CD25+FOXP3+.
  • Line 336: Panels showing nTreg parameters in Figure 2 were moved into Supplementary figure 4 for the sake of clearer data presentation. Correspondingly, figure caption of Supplementary figure 4 was updated in the Supplementary files.
  • Lines 337-353: Figure caption was updated to reflect the removal of panels showing nTreg. All relevant references to Supplementary figures were re-numbered. All instances of CD25highFOXP3+ were substituted with CD25+FOXP3+.
  • Lines 359-366: All instances of CD25highFOXP3+ were substituted with CD25+FOXP3+.
  • Lines 370-371: Added to improve the clarity of the presented data.
  • Line 375: Relevant supplementary figures have been re-numbered.
  • Lines 376-377: In response to a suggestion offered by Reviewer 1, Pearson’s correlation coefficients and p-values of all the correlated parameters were summarised in Supplementary tables 1 and 2. This was stated in the added sentence.
  • Lines 380-403: In response to a suggestion made by Reviewer 1, graphical presentation of correlation data was reorganised and the less relevant correlations were moved to Supplementary figures 7 through 9. This was done in the hope of improving the clarity of data representation. Figure and panel captions were updated to reflect the removal of panels and renumbering of the Supplementary figures.
  • Lines 404-405: Reference to Supplementary table 1 was added.
  • Lines 408-425: Removal of less relevant correlations from the main body of the article into Supplementary figure 9, as per the reviewers’ suggestion. Figure and panel captions were updated to reflect the removal of panels and renumbering of the Supplementary figures.
  • Lines 426-428: Reference to Supplementary table 2 was added.
  • Lines 429-437: Gene expression analysis of Il10, Tgfb and Il35 (p35 subunit) on mRNA level was added to the supplementary files (Supplementary figure 11). Introduction of the findings added into the Results section of the manuscript.
  • Line 448: IL-1 was changed to more precise IL-1β (also changed in the relevant Supplementary figure 10). Supplementary figure number updated.
  • Lines 451-454: Gene expression analysis of Il10, Tgfb and Il35 (p35 subunit) on mRNA level was added to the supplementary files (Supplementary figure 11). Short description of the findings was added.
  • Line 462: Supplementary figure number updated.
  • Line 466: Figure 4 was reorganised in order to improve clarity and to present the most relevant findings. Pearson’s correlation coefficients (r) and p-values were included in all shown panels, as suggested by both reviewers. Less relevant panels were moved into Supplementary figures 7 and 8.
  • Lines 470-482: Figure caption was updated to reflect the reorganization of figures. References to individual figures and panels were accordingly re-labelled. All instances of CD25highFOXP3+ were substituted with CD25+FOXP3+.
  • Line 486: Figure 5 was reorganised in order to improve clarity and to present the most relevant findings. Pearson’s correlation coefficients (r) and p-values were included in all shown panels, as suggested by both reviewers. Less relevant panels were moved into Supplementary figure 9.
  • Lines 487-497: Figure caption was updated to reflect the reorganization of figures. References to individual figures and panels were accordingly re-labelled.
  • Line 523: An instance of CD25highFOXP3+ was substituted with CD25+FOXP3+.
  • Lines 534-547: Caesarean section may represent a biologically highly relevant condition, potentially contributing to the discrepancy between the results presented in this study and the observations previously reported by our group, which were made on cord blood of children delivered vaginally. Discussion of the potential influence of caesarean section on the immunological parameters in cord blood, including relevant literature references, was therefore added. This includes references reporting increased risk of allergic diseases in children delivered by caesarean section. The cited data suggests a potential confounding effect of caesarean section on immune regulation, which may represent one of the factors responsible for the onconsistency between the current study and previous studies published by our group.
  • Lines 559-562: Added in order to clearly state the desirability of introducing a widely accepted methodology for regulatory T cell study.
  • Lines 593-611: More rigorous and more substantially referenced discussion of the increased risk of various childhood allergic manifestations in male children was added in response to a comment by Reviewer 1.
  • Lines 616-625: A paragraph added to address a comment by Reviewer 1 by summarising the significance of our findings regarding Helios+ (putative nTreg) population in cord blood of male newborns. We also acknowledge the need for more extensive studies to conclusively determine the biological significance of the reported phenomenon.
  • Lines 637-641: Correction of several minor typographical or stylistic errors.
  • Line 658: An instance of CD25highFOXP3+ was substituted with CD25+FOXP3+.
  • Line 682: Gene expression analysis of regulatory added to the list of methods usable for indirect analysis of Treg function.
  • Line 694: IL-1 was changed to more precise IL-1β.
  • Lines 697-698: Mention was made of results of gene expression analysis of Il10, Tgfb and Il35.
  • Lines 706-711: Discussion of the role of regulatory IL-35 in control of allergic was added, including references to published studies.
  • Lines 713-714: Low number of included samples was clearly stated as the major limitation of the functional assays presented in our manuscript.
  • Line 716: Minor stylistic change.
  • Line 724: An instance of CD25highFOXP3+ was substituted with CD25+FOXP3+.
  • Lines 746-747: Stressed the biological significance of clinical factors such as caesarean section.
  • Lines 750-755: Slightly expanded the section discussing the functional context of the observed difference between Helios+ Treg in male and female newborns, including more relevant references to literature as well as a clear admission that further studies are needed to fully elucidate the significance of our finding.
  • Line 756, line 761: Minor stylistic changes.
  • Lines 763-786: Added a paragraph openly stating and discussing the limitations of our study in order to address some concerns expressed by the reviewers. We stressed the need for an establishment of a generally accepted consensus for study of regulatory T cells and their subpopulations, as well as the higher usefulness of functional assays as opposed to solely proportion-based analysis of Treg subpopulations. We pointed out the main finding of presented data, namely slightly higher Helios+ nTreg subpopulation in male children, and stated the need of further, well performed studies to evaluate the biological significance of this finding.
  • Lines 798-815: Updated listing of the supplementary figures and tables, which were extensively altered due to the changes in data presentation in the main body of the manuscript.
  • Lines 829-830: Correction of minor stylistic errors.

Responses to reviewer 1 comments

Point 1: The difference that represents the main (only) finding of the study is minimal, and flow cytometry data are not shown except for a representative group of dot plots showing the gating strategy as a supplemental figure.  An example of the difference in MFI (Helios) between groups should be shown by a histogram plot.

We have attempted to reorganise the shown gating strategy to be more comprehensive. As requested, we have included representative histograms showing difference in MFI of Helios between the various groups compared in our study. We have also included fluorescence minus one (FMO) control for Helios. To further improve the gating strategy we’re showing, we have additionally included representative histograms of MFI of FOXP3 for cord blood and maternal peripheral blood (the two groups showing significant difference), including FMO control for FOXP3. We hope that this gating strategy is more concise and better supports relevance of the flow cytometric data we are presenting in the manuscript.

We have elected to include the entire gating strategy in a single figure to show the overview of the entire workflow of flow cytometric data analysis. As an alternative, the gating strategy might be divided into two figures, with dot plots showing the selection of populations down to the CD25+FOXP3+ Treg in one figure and with the representative histograms of Helios and FOXP3 MFI as the second figure. Should you deem this proposed alternative preferable, we would be happy to implement it in another round of revisions.

Point 2: Another relevant concern regards the limitation of the functional analysis. Cytokine secretion in T conv stimulated with anti-CD3 /CD28 is detected only in CB from children from healthy mother (only IL-10 and TNF) and no suppression or significant variation is observed upon addition of Tregs.

We admit the limitations of the functional analysis shown in our manuscript. This is caused mainly by the low number of probands included for functional studies, and we have added statements to that effect into the discussion (lines 713; 778-779). While low number of samples analysed is the most important reason of the insufficient statistical power, it should also be noted that human samples in general and human cord blood samples in particular are known to possess high interindividual variability. In case of cord blood, this is reflective of the highly abnormal conditions accompanying delivery, which differ depending on various factors such as mode of delivery. Children in our study were born by caesarean section; accordingly, we have added a paragraph discussing the effects this form of birth may exert on neonatal immune system, including cord blood (lines 534-547). Another important factor to consider is that cord blood immune cells are generally less mature, as evidenced e.g. by the lower FOXP3 MFI compared to maternal peripheral blood; Treg in cord blood are thus conceivably less functionally capable, which might explain the lack of pronounced suppressive effect on TNF-α as well as the fact that no significant increment in regulatory cytokine production upon Treg addition to cell cultures was seen in this study.

Furthermore, as an alternative approach to analysing functional competency of cord blood Treg, we have newly included results of gene expression analysis of mRNA encoding regulatory cytokines IL‑10, TGF-β and IL-35 in magnetically isolated cord blood Treg (Supplementary figure 11). Although statistical significance was not reached, again likely due to the low number of samples analysed, there is a discernible trend towards lower expression of regulatory cytokines in children of allergic mothers, consistent with the trends we have described in the coculture-based assays.

Point 3: The evidence of a different percentage of iTreg and nTreg in male vs female newborns is novel but is not really discussed. No previous studies supporting this evidence are included in the reference list and the references cited in support to an increased male tendency to develop allergies at an early age are few and limited in their impact.

We have expanded our discussion of the observed sex-dependent difference in iTreg and nTreg in the main body of the manuscript. To our best knowledge, no study has been published including comparing Helios and proportion of Helios-expressing Treg cells; we have included this statement into the main body of the article (lines 616-626) and also offered some ideas for further studies which could provide further corroboration and conclusively determine whether this finding is biologically significant or is in fact incidental.

We have also added further references supporting our claim of increased incidence of a wide variety of childhood allergies in males, including analyses performed in different cohorts and on different genetic backgrounds, as well as meta-analyses (lines 598-610). Hopefully, the offered evidence shows that the described effects are to a substantial degree caused by intrinsic sex-dependent factors rather than environmental causes and supports our claims acceptably.

Point 4: There are differences with previous findings mentioned in the discussion that the authors ascribe to differences in technical protocols and methodological approaches. This is a major limitation. Therefore it is necessary to support the new evidence (male vs female Treg) with very clear data accurately describing and showing data analysis.

We fully agree that consistent technical protocols and methodological approaches are of paramount importance when performing flow cytometry analyses; we have in fact noted that establishing a generally accepted and utilized consensus for Treg analysis would be highly useful for achieving better comparability between studies performed by different groups and in different settings. We hope that the improved gating strategy shown in Supplementary figure 1, including representative histograms and FMO controls for intracellular markers FOXP3 and Helios, will serve as a sufficiently persuasive proof of the relevance of the flow cytometry data presented in our study.

Furthermore, we also believe that the reported differences between this study and previous studies published by our lab may be at least partially due to the difference in the studied cohort. In this case, children were born via caesarean section as opposed to the vaginal delivery of children from cohorts reported in our older studies, representing notably different biological settings. We have added a paragraph discussing the relevance of mode of delivery for immune system characteristics (lines 543-547). While so far there is insufficient amount of reliable studies regarding the effect of caesarean section on Treg, the reported effects on immune system and neonatal immune cells in general may at least partially account for the different results described in this study.

Point 5: Figures are difficult to follow and should be reorganized for a clearer data representation.

We have attempted to reorganize the majority of figures to comply with this suggestion. Firstly, we have moved all panels showing Helios+ nTreg into the Supplementary files, since they by definition mirror the trends of Helios- iTreg. Furthermore, we reorganized the data showing correlation analysis according to the suggestions made in Point 6, limiting the correlations shown to the most relevant findings and making the graphs hopefully clearer and more self-contained.

If you have any specific suggestions for how we could further make the graphical representation of data clearer, we will be happy to comply with these in another round of revisions.

Point 6: Many correlation analyses are reported, some of them not really adding new information. This part could be substituted with a table clearly reporting r and p values for each correlation. (As an alternative, interpolating lines together with the value of r and p for each correlation should be included in individual plots. Only the most relevant findings should be shown).

We have moved the less relevant correlations into Supplementary figures and attempted to make all correlation plots clearer by adding the interpolation lines for the statistically significant correlations, as well as by including the correlation coefficients and p values in each individual scatter plot.

Furthermore, we have also included tables reporting r and p values for all calculated correlation analyses as Supplementary tables 1 and 2 to provide a clear overview of the reported correlations.

Point 7: Tregs are defined as CD25hi FOXp3+ but the only dot plot reported in supplemental data shows mainly CD25+ FOXp3+ CD4+ T cells. Please clarify.

We thank you for this comment, as indeed it is more accurate to define Treg in the data presented in this study as CD25+FOXP3+ rather than CD25highFOXP3+ cells. We have substituted CD25+FOXP3+ for CD25highFOXP3+ in the entire article.

Reviewer 2 Report

Dear Authors:

The manuscript entitled “Lower functional and proportional characteristics of cord blood Treg of male newborns compared with female newborns” by Cerny et al. is very well written and contains relevant new information concerning Treg cells in infants born to healthy mothers versus mothers with allergies.

The aim of the study was to obtain information that would assist in the early treatment of allergies in children born to mothers with allergies. To this end, Treg cells present in cord blood were separated into iTreg and nTreg by flow cytometry via staining with Helios, which is a marker for nTreg cells. The data indicates that iTreg cells are higher in the female infants than in the males and nTreg cells are lower in the females than in the male children from healthy mothers. Data for infants of mothers with allergies was not significant. The MFI of FOXP3 was found to be higher in male infants born to healthy mothers. Cytokines in media from Treg and Tconv cocultures did not provide significant differences between stimulated and stimulated 1:5 Treg:Target for the cytokines measured. The authors indicate that the data suggest “that immaturity of neonatal immune system is more severe in males, predisposing them to increased risk of allergy development”.

Major Points

  1. The sample of the flow cytometry gating in the supplemental section appears adequate, but more accurate results could be obtained if better separation of FOXP3- and FOXP3+ cells and Helios- and Helios+ would be possible to achieve.
  2. Significant differences in cord Treg populations between male and female infants are being reported primarily for infants born to healthy mothers.   Isn’t the risk for developing allergies higher in children born to mothers with allergies? However, the data does not indicate strong differences in males and females from mothers with allergies.
  3. The cell culture cytokine results do show a trend toward decreased regulatory function of Treg cells obtained from cord blood of infants born to mothers with allergies. It could be pointed out, too, that there is slightly decreased cytokine production from the stimulated Tconv cells form cord blood of infants from mothers with allergies compared to cells obtained from cord blood of healthy mothers. Although the standard errors are too high to allow for significance (which is unfortunate), perhaps some significant differences can be obtained by comparing results from data based on healthy mothers versus data based on mothers with allergies. This could possibly be done by calculating percent differences in stimulation and regulation between healthy mothers and mothers with allergies.
  4. Scatter plots presented to show correlations, or lack thereof, do not have regression lines. It would be easier for the readers to see the correlations if regression lines were added to these figures. This is just a suggestion.
  5. The conclusion that the Treg cells present in male infant cord blood are more immature and less functional than those in the cord blood of females is not strongly supported by the data, since significant differences between males and females were mostly detected in cord blood from healthy mothers. Therefore, indicate that this conclusion is suggested by the data, but more studies are needed to obtain data to support this hypothesis.

Author Response

Firstly, we sincerely thank the editor and the reviewers for the time taken to read our manuscript and for formulating the comments, suggestions and critiques. We appreciate the positive comments and offered suggestions. We endeavoured to revise the manuscript to address the extensive and insightful concerns of the reviewers.

Point-to-point listing of all revisions made to the manuscript:

All changes are listed by appearance from the beginning to the end of the text. Lines containing the major changes to the revised manuscript are shown in bold for better orientation. Lines provided correspond to the numbering in the document showing all changes (i.e. the tracked copy of the revised manuscript).

  • Line 47: As a response to a comment of Reviewer 1, we substituted all instances of CD25highFOXP3+ in our text with the more correct CD25+FOXP3+. First example of this substitution can be found on the line number 47; the substitutions were consistently made in the body of the entire article as well as in figure captions and in the Supplementary files. The substitutions number 23 in the main article and 13 in the Supplementary files.
  • Lines 141-150: In response to a comment of Reviewer 1, we expanded our gating strategy presented in Supplementary figure 1 and added representative histograms of FOXP3 and Helios fluorescence intensity, showing differences between the various groups which we report in the main body of the article. The description of the gating strategy has been correspondingly expanded, as seen on lines 141-150. Captions of various panels were likewise changed to reflect the definitive panels included in Supplementary figure 1. Correspondingly, figure caption of Supplementary figure 1 was updated in the Supplementary files.
  • Lines 161-164: Deleted to maintain constant spacing between the paragraphs in the article.
  • Lines 165-188: To further support our functional assays, data regarding gene expression of Il10, Tgfb and Il35 (p35 subunit) on mRNA level was added to the supplementary files (Supplementary figure 11). Description of methodical aspects analysis was added to the Materials and Methods section of main body of the article, lines 165 through 188.
  • Line 224: FTS was substituted with the more correct foetal calf serum.
  • Line 228: State and address of the provider were removed since they are now listed upon the first occurrence on the added line 184.
  • Line 234: IL-1 was changed to more precise IL-1β.
  • Lines 260-261: A sentence describing the added gene expression analysis of regulatory cytokines was added.
  • Lines 264-269: Added to explain the changes made to figures 1 and 2 in response to the comment of Reviewer 1, which suggests that the figures should be reorganized and made more comprehensive for the reader. To comply, we moved the panels showing Helios+ nTreg population characteristics into newly included Supplementary figures 2 and 4. This change in graphical data presentation and inclusion of panels in the main body of the article and in the Supplementary files resulted in the necessity of re-labelling all subsequent references to Supplementary figure numbers, as well as references to the panels in Figures 1 and 2 in the main body of the article, starting on line 273.
  • Lines 273 through 317: References to individual figures and panels were re-labelled according to the reorganization of figures, as mentioned previously.
  • Line 299: Since the abbreviation MFI was introduced earlier, explanation of the abbreviation was removed for the sake of consistency.
  • Line 318: Panels showing nTreg parameters in Figure 1 were moved into Supplementary figure 2 for the sake of clearer data presentation. Correspondingly, figure caption of Supplementary figure 2 was updated in the Supplementary files.
  • Lines 320-335: Figure caption was updated to reflect the removal of panels showing nTreg. All relevant references to Supplementary figures were re-numbered. All instances of CD25highFOXP3+ were substituted with CD25+FOXP3+.
  • Line 336: Panels showing nTreg parameters in Figure 2 were moved into Supplementary figure 4 for the sake of clearer data presentation. Correspondingly, figure caption of Supplementary figure 4 was updated in the Supplementary files.
  • Lines 337-353: Figure caption was updated to reflect the removal of panels showing nTreg. All relevant references to Supplementary figures were re-numbered. All instances of CD25highFOXP3+ were substituted with CD25+FOXP3+.
  • Lines 359-366: All instances of CD25highFOXP3+ were substituted with CD25+FOXP3+.
  • Lines 370-371: Added to improve the clarity of the presented data.
  • Line 375: Relevant supplementary figures have been re-numbered.
  • Lines 376-377: In response to a suggestion offered by Reviewer 1, Pearson’s correlation coefficients and p-values of all the correlated parameters were summarised in Supplementary tables 1 and 2. This was stated in the added sentence.
  • Lines 380-403: In response to a suggestion made by Reviewer 1, graphical presentation of correlation data was reorganised and the less relevant correlations were moved to Supplementary figures 7 through 9. This was done in the hope of improving the clarity of data representation. Figure and panel captions were updated to reflect the removal of panels and renumbering of the Supplementary figures.
  • Lines 404-405: Reference to Supplementary table 1 was added.
  • Lines 408-425: Removal of less relevant correlations from the main body of the article into Supplementary figure 9, as per the reviewers’ suggestion. Figure and panel captions were updated to reflect the removal of panels and renumbering of the Supplementary figures.
  • Lines 426-428: Reference to Supplementary table 2 was added.
  • Lines 429-437: Gene expression analysis of Il10, Tgfb and Il35 (p35 subunit) on mRNA level was added to the supplementary files (Supplementary figure 11). Introduction of the findings added into the Results section of the manuscript.
  • Line 448: IL-1 was changed to more precise IL-1β (also changed in the relevant Supplementary figure 10). Supplementary figure number updated.
  • Lines 451-454: Gene expression analysis of Il10, Tgfb and Il35 (p35 subunit) on mRNA level was added to the supplementary files (Supplementary figure 11). Short description of the findings was added.
  • Line 462: Supplementary figure number updated.
  • Line 466: Figure 4 was reorganised in order to improve clarity and to present the most relevant findings. Pearson’s correlation coefficients (r) and p-values were included in all shown panels, as suggested by both reviewers. Less relevant panels were moved into Supplementary figures 7 and 8.
  • Lines 470-482: Figure caption was updated to reflect the reorganization of figures. References to individual figures and panels were accordingly re-labelled. All instances of CD25highFOXP3+ were substituted with CD25+FOXP3+.
  • Line 486: Figure 5 was reorganised in order to improve clarity and to present the most relevant findings. Pearson’s correlation coefficients (r) and p-values were included in all shown panels, as suggested by both reviewers. Less relevant panels were moved into Supplementary figure 9.
  • Lines 487-497: Figure caption was updated to reflect the reorganization of figures. References to individual figures and panels were accordingly re-labelled.
  • Line 523: An instance of CD25highFOXP3+ was substituted with CD25+FOXP3+.
  • Lines 534-547: Caesarean section may represent a biologically highly relevant condition, potentially contributing to the discrepancy between the results presented in this study and the observations previously reported by our group, which were made on cord blood of children delivered vaginally. Discussion of the potential influence of caesarean section on the immunological parameters in cord blood, including relevant literature references, was therefore added. This includes references reporting increased risk of allergic diseases in children delivered by caesarean section. The cited data suggests a potential confounding effect of caesarean section on immune regulation, which may represent one of the factors responsible for the onconsistency between the current study and previous studies published by our group.
  • Lines 559-562: Added in order to clearly state the desirability of introducing a widely accepted methodology for regulatory T cell study.
  • Lines 593-611: More rigorous and more substantially referenced discussion of the increased risk of various childhood allergic manifestations in male children was added in response to a comment by Reviewer 1.
  • Lines 616-625: A paragraph added to address a comment by Reviewer 1 by summarising the significance of our findings regarding Helios+ (putative nTreg) population in cord blood of male newborns. We also acknowledge the need for more extensive studies to conclusively determine the biological significance of the reported phenomenon.
  • Lines 637-641: Correction of several minor typographical or stylistic errors.
  • Line 658: An instance of CD25highFOXP3+ was substituted with CD25+FOXP3+.
  • Line 682: Gene expression analysis of regulatory added to the list of methods usable for indirect analysis of Treg function.
  • Line 694: IL-1 was changed to more precise IL-1β.
  • Lines 697-698: Mention was made of results of gene expression analysis of Il10, Tgfb and Il35.
  • Lines 706-711: Discussion of the role of regulatory IL-35 in control of allergic was added, including references to published studies.
  • Lines 713-714: Low number of included samples was clearly stated as the major limitation of the functional assays presented in our manuscript.
  • Line 716: Minor stylistic change.
  • Line 724: An instance of CD25highFOXP3+ was substituted with CD25+FOXP3+.
  • Lines 746-747: Stressed the biological significance of clinical factors such as caesarean section.
  • Lines 750-755: Slightly expanded the section discussing the functional context of the observed difference between Helios+ Treg in male and female newborns, including more relevant references to literature as well as a clear admission that further studies are needed to fully elucidate the significance of our finding.
  • Line 756, line 761: Minor stylistic changes.
  • Lines 763-786: Added a paragraph openly stating and discussing the limitations of our study in order to address some concerns expressed by the reviewers. We stressed the need for an establishment of a generally accepted consensus for study of regulatory T cells and their subpopulations, as well as the higher usefulness of functional assays as opposed to solely proportion-based analysis of Treg subpopulations. We pointed out the main finding of presented data, namely slightly higher Helios+ nTreg subpopulation in male children, and stated the need of further, well performed studies to evaluate the biological significance of this finding.
  • Lines 798-815: Updated listing of the supplementary figures and tables, which were extensively altered due to the changes in data presentation in the main body of the manuscript.
  • Lines 829-830: Correction of minor stylistic errors.

Responses to reviewer 2 comments

Point 1: The sample of the flow cytometry gating in the supplemental section appears adequate, but more accurate results could be obtained if better separation of FOXP3- and FOXP3+ cells and Helios- and Helios+ would be possible to achieve.

We thank you for the positive commentary. We agree that robust separation of positive and negative samples is crucial for accurate and responsible analysis of flow cytometry data, even though this is sometimes tricky to achieve with less robustly expressed markers, particularly intracellular proteins including transcription factors and cytokines. In order to better show the differences between Helios+ and Helios- populations in the various groups compared, as well as the differences in FOXP3 MFI between maternal and cord blood Treg, we have included an expanded and hopefully improved gating strategy, which shows representative histograms of FOXP3 in cord blood and maternal peripheral blood Tregs, as well as representative histograms of Helios in the various pairings of samples which we have compared in the article. Fluorescence minus one (FMO) controls for FOXP3 and Helios were also included.

Point 2: Significant differences in cord Treg populations between male and female infants are being reported primarily for infants born to healthy mothers.   Isn’t the risk for developing allergies higher in children born to mothers with allergies? However, the data does not indicate strong differences in males and females from mothers with allergies.

You are correct that children of allergic mothers are generally considered to have higher risk of allergy; as such, the lack of a significant difference between male and female children in this higher-risk group is somewhat surprising. Various confounding factors may contribute to this perceived inconsistency. Firstly, it bears mentioning that percentage of Helios+ Treg and Helios MFI levels represent only one aspect of the rather complex interplay of immune-relevant intrinsic and extrinsic factors which stochastically combine into the actual risk of allergy development. In particular, delivery by caesarean section is itself a predisposing factor for allergy development, as mentioned in the discussion (lines 534-547). The various risk factors are also unlikely to have the same weight and predictive value for allergy development, particularly when they are present in combinations. We suspect that although male sex confers a slight increase in risk of allergy, this is overshadowed by the stronger risk factors, including allergy status and caesarean section. Since all children in our study were born by caesarean section, the allergic cohort in fact is already burdened by two of the presumably stronger risk factors for allergy development; this may make it harder to reach a statistically significant result in this group, although the trend is in fact similar as in the group of children of healthy mothers.

Furthermore, the exact biological role of Helios transcription factor itself and of Helios+ Treg has not been fully and conclusively determined, and the identification of induced and natural Treg is a notoriously contentious issue. It may thus be conceivable that Helios+ (Helios-) cells may in fact represent a mixture of nTreg (iTreg) cells contaminated by other T cell populations; the exact composition and percentage of such contaminating cells may also vary depending on the other risk factors which are present (e.g. allergic parentage, caesarean section etc.). It is therefore conceivable that these additional risk factors may further mask the risk conferred on male children by their sex by contaminating Helios- induced Treg e.g. with a small proportion of activated non-Treg T cells which have recently upregulated CD25 and FOXP3, both of which have been implicated as markers of Tcell activation. Such contaminating activated Tconv subpopulation might conceivably be similar in male and in female children, and may represent a mechanism by which a risk factor of lesser functional significance, i.e. sex, might be overshadowed by the larger ones. Thus, the sex-dependent difference between bona fide Helios- Treg may be easier to determine in the cohort which is burdened by fewer of the stronger risk factors, i.e. in children of healthy mothers in the case of our study.

Point 3: The cell culture cytokine results do show a trend toward decreased regulatory function of Treg cells obtained from cord blood of infants born to mothers with allergies. It could be pointed out, too, that there is slightly decreased cytokine production from the stimulated Tconv cells form cord blood of infants from mothers with allergies compared to cells obtained from cord blood of healthy mothers. Although the standard errors are too high to allow for significance (which is unfortunate), perhaps some significant differences can be obtained by comparing results from data based on healthy mothers versus data based on mothers with allergies. This could possibly be done by calculating percent differences in stimulation and regulation between healthy mothers and mothers with allergies.

We agree that the low number of included probands, as well as the inherently high interindividual variability of cord blood samples, represent a significant limitation of the reported functional assays; we have endeavoured to state this pitfall more openly in the discussion (lines 713; 778-779). Another potential factor contributing to the observed variability might be the mode of delivery of children included in this study, i.e. caesarean section, which constitutes a highly artificial setting, one which is currently insufficiently understood as far as regulatory T cells are concerned.

As you have suggested, we have attempted to calculate stimulation indexes (defined as ratios of cytokine concentration in stimulated samples to concentration in unstimulated control samples) as well as assess regulation (i.e. percent decrease of TNF-α production in samples with added Treg) and stimulation (i.e. percent increase in IL-10 and TGF-β production in samples with added Treg). Unfortunately, no significant difference between any of these parameters was uncovered when Treg isolated from cord blood of children of allergic and healthy mothers were compared.

As an alternative approach to analysing functional competency of cord blood Treg, we have newly included results of gene expression analysis of mRNA encoding regulatory cytokines IL‑10, TGF-β and IL-35 in magnetically isolated cord blood Treg (Supplementary figure 11). Although statistical significance was not reached, again likely due to the low number of samples analysed, there is a discernible trend towards lower expression of regulatory cytokines in children of allergic mothers, consistent with the trends we have described in the coculture-based assays.

Point 4: Scatter plots presented to show correlations, or lack thereof, do not have regression lines. It would be easier for the readers to see the correlations if regression lines were added to these figures. This is just a suggestion.

We thank you for this useful suggestion. We have added regression lines to all scatter plots showing significant correlation, both in the main body of the article and in the Supplementary files. In compliance with suggestion made by the other reviewer and in order to improve the comprehensiveness of the presented data, we have also added correlation coefficients and p values into each scatter plot, moved the panels showing less significant correlations from the main body into the Supplementary figures and included two supplementary tables showing a comprehensive overview of correlation coefficients and p values of all correlated values.

Point 5: The conclusion that the Treg cells present in male infant cord blood are more immature and less functional than those in the cord blood of females is not strongly supported by the data, since significant differences between males and females were mostly detected in cord blood from healthy mothers. Therefore, indicate that this conclusion is suggested by the data, but more studies are needed to obtain data to support this hypothesis.

We have reformulated the relevant parts of discussion in accordance with this suggestion and stated the need for further, more detailed studies into this phenomenon (lines 568; 616-626; 775-777). Nevertheless, as we have likewise stated in the discussion, to the best of our knowledge, our findings represent the first reported study of sex differences of Helios expression in Treg in the context of cord blood, making further studies into the phenomenon that much more important.

If the revised formulations still seem unwarrantedly strong to you, please let us know so that further and more stringent discussion may be added in another round of revisions.

Round 2

Reviewer 1 Report

The major points have been addressed. The discussion now though being more comprehensive is about 5 printed pages and needs to be shortened. 

Author Response

We would like to thank the reviewers for giving us the opportunity to submit a revised version of our manuscript to special issue of the journal Biomedicines. We thank the reviewers for the time they spent to review the revised version of our manuscript and for their positive feedback. We are submitting the revised the manuscript to address the suggestion for shortening the discussion section. We have tried to keep the major points discussed in revised version of the manuscript while eliminating repetitive sections and reducing verbosity. The current length of the revised discussion corresponds to the length of discussion of the originally submitted manuscript. We believe that the length of discussion section is now appropriate while retaining the key points, offering proper context and remaining comprehensive. Nevertheless, we will be happy to review or remove any additional parts of discussion deemed unnecessary and/or onerous based on reviewer´s recommendation.

The changes we have made to the manuscript in this round of revision are highlighted by blue colour.